Genome-wide identification and transcriptional expression analysis of superoxide dismutase (SOD) family in wheat (Triticum aestivum)

Jiang Wenqiang 1 2 3
Yang Lei 1
He Yiqin 1
Zhang Haotian 1
Li Wei 3
Chen Huaigu 3
Ma Dongfang madf@yangtzeu.edu.cn 1 2
Yin Junliang yinjunliang@nwafu.edu.cn 1
1 Engineering Research Center of Ecology and Agricultural Use of Wetland, Ministry of Education/Hubei Collaborative Innovation Center for Grain Industry/College of Agriculture, Yangtze University , Jingzhou , Hubei , China
2 Institute of Plant Protection and Soil Science, Hubei Academy of Agricultural Sciences , Wuhan , Hubei , China
3 Institute of Plant Protection, Jiangsu Academy of Agricultural Sciences , Nanjing , Jiangsu , China
Lazo Gerard
Electronic publication date: 2019 Nov 19
Publication date: 2019
Volume: 7
Electronic Location ID: e8062
Received 2018 Nov 29; Accepted 2019 Oct 20
Copyright: ©2019 Jiang et al.
Copyright year: 2019
Copyright holder: Jiang et al.
License: This is an open access article distributed under the terms of the Creative Commons Attribution License, which permits unrestricted use, distribution, reproduction and adaptation in any medium and for any purpose provided that it is properly attributed. For attribution, the original author(s), title, publication source (PeerJ) and either DOI or URL of the article must be cited.
License URL: https://creativecommons.org/licenses/by/4.0/

Keywords: SOD, Gene structure, Protein characterization, Abiotic stress, Expression profiles

Funding: National Key R&D Program of China 2018YFD0200500 Open Project Program of State Key Laboratory for Biology of Plant Disease and Insect Pests SKLOF201707 Open Project Program of Engineering Research Center of Ecology and Agricultural Use of Wetland, Ministry of Education KF201802 This work was supported by the “National Key R&D Program of China (2018YFD0200500)”, “Open Project Program of State Key Laboratory for Biology of Plant Disease and Insect Pests (SKLOF201707)” and the “Open Project Program of Engineering Research Center of Ecology and Agricultural Use of Wetland, Ministry of Education (KF201802)”. The funders had no role in study design, data collection and analysis, decision to publish, or preparation of the manuscript.

==============================
Superoxide dismutases (SODs) are a family of key antioxidant enzymes that play a crucial role in plant growth and development. Previously, this gene family has been investigated in Arabidopsis and rice. In the present study, a genome-wide analysis of the SOD gene family in wheat were performed. Twenty-six SOD genes were identified from the whole genome of wheat, including 17 Cu/Zn-SODs, six Fe-SODs, and three Mn-SODs. The chromosomal location mapping analysis indicated that these three types of SOD genes were only distributed on 2, 4, and 7 chromosomes, respectively. Phylogenetic analyses of wheat SODs and several other species revealed that these SOD proteins can be assigned to two major categories. SOD1 mainly comprises of Cu/Zn-SODs, and SOD2 mainly comprises of Fe-SODs and Mn-SODs. Gene structure and motif analyses indicated that most of the SOD genes showed a relatively conserved exon/intron arrangement and motif composition. Analyses of transcriptional data indicated that most of the wheat SOD genes were expressed in almost all of the examined tissues and had important functions in abiotic stress resistance. Finally, quantitative real-time polymerase chain reaction (qRT-PCR) analysis was used to reveal the regulating roles of wheat SOD gene family in response to NaCl, mannitol, and polyethylene glycol stresses. qRT-PCR showed that eight randomly selected genes with relatively high expression levels responded to all three stresses based on released transcriptome data. However, their degree of response and response patterns were different. Interestingly, among these genes, TaSOD1.7, TaSOD1.9, TaSOD2.1, and TaSOD2.3 feature research value owing to their remarkable expression-fold change in leaves or roots under different stresses. Overall, our results provide a basis of further functional research on the SOD gene family in wheat and facilitate their potential use for applications in the genetic improvement on wheat in drought and salt stress environments.

Introduction

During the growth process, plants are typically affected by various adverse factors (such as drought, water damage, heat damage, cold damage, pests and pathogens, heavy metal ions, etc.), which lead to a production of large amounts of reactive oxygen species (ROS) in plants (Razali et al., 2015). ROS accumulation causes oxidative stress, which destroys biological macromolecules, biofilms, and other structures, and in severe cases causes cell death (Foyer & Noctor, 2005; Quan et al., 2010). However, ROS as signal molecules, regulate many physiological processes during plant growth and development, and participate in various biotic and abiotic stress responses (Mittler, 2002; Pitzschke, Forzani & Hirt, 2006). Plants have evolved complex antioxidant enzyme system that inhibits ROS accumulation, which is mediated predominantly by superoxide dismutase (SOD), catalase (CAT), peroxidase (POD), ascorbic acid (AsA), glutathione (GH), and ascorbate peroxidase (APX) (Alscher, Erturk & Heath, 2002; Valko et al., 2006; Sugimoto et al., 2014; Zhang et al., 2016c). Increased stress resistance in plants may be related to the organism’s antioxidant enzyme system in the body (Guo et al., 2017). SOD is widely present in living organisms. As the first enzyme involved in the scavenging reaction of reactive oxygen species, SOD is involved in almost most physiological and biochemical reactions against various environmental stressors in organisms, and is at the core of antioxidant enzymes (Song et al., 2009; Ahmad, Umar & Sharma, 2010; Dong et al., 2013). Fridovieh and Mccor (1969) first revealed the biological function of SODs that can catalyze the conversion of superoxide (O2−) into oxygen (O2) and hydrogen peroxide (H2O2) through disproportionation, and further convert H2O2 into water (H2O) by peroxidase and oxidase enzymes to achieve active oxygen removal (Tepperman & Dunsmuir, 1990). SOD plays an important role in scavenging oxygen-free radicals, thereby preventing oxygen-free radicals from disrupting the composition, structure, and function of cells and protecting cells from oxidative damage (Ding, 2008).

Many plants contain a series of SOD isozymes and the belongs to metalloproteinases, which their proteins are catalytically active after obtaining metal prosthetic groups (Bowler, Montagu & Inze, 1992). According to the different metal cofactors in the catalytic site, it can be divided into four types: Cu/Zn-SOD, Mn-SOD, Fe-SOD, and Ni-SOD (Abreu & Cabelli, 2010; Whittaker, 2010). Fe-SOD and Mn-SOD occur mainly present in lower plants, whereas Cu/Zn-SOD is mainly found in higher plants (Xia et al., 2015; Zeng et al., 2014). Ni-SOD is found in Streptomyces, cyanobacteria and marine life (Kim et al., 1996; Wuerges et al., 2004; Dupont et al., 2010). These SODs were distributed in different regions of the cell and play a key role in responding to oxidative stress (Miller, 2004). Previous studies found that Fe-SOD occurs in chloroplasts; Mn-SOD in mitochondria and peroxisomes; Cu/Zn-SOD mainly occurs in chloroplasts and in the cytoplasm and Ni-SOD mainly exists in the cytoplasm (Youn et al., 1996; Dupont et al., 2010).

Numerous studies have shown that the expression of plant SOD genes are affected by various environmental stressors, and different environmental conditions lead to differences in SOD gene expression (Xia et al., 2015; Zhang et al., 2016c). SOD activity in Oryza sativa (Lin et al., 2009) and Pisum sativum (Yan et al., 2009) was increased under salt stress. In arid environments, the activity of SOD decreased in Arachis hypogaea at the early stage of stress, but under severe drought stress, SOD activity increased (Jiang & Ren, 2004). Compared to warmer temperatures, at 4 °C, the Cu/Zn-SOD activity in barley leaves did not change significantly; however the temperature dropped to −3 °C, Cu/Zn-SOD activity increased significantly (Moses, 2012). Under drought and saline conditions, higher drought resistance and salt tolerance in transgenic plants with the AtHDG11 gene increased, whereas SOD activity increased, suggesting the importance of SOD for plant resistance. When the Arabidopsis CBF1 (C-repeat-binding factor 1) gene was transferred to tobacco, SOD activity in transgenic tobacco plants was significantly higher than that in controls, which improved the tolerance of transgenic plants to low temperatures (Zhang et al., 2010). Over-expression of Mn-SOD in tobacco and maize chloroplasts enhanced the protective effect on the plasma membrane in transgenic tobacco and maize and increased tolerance to herbicide-induced oxygen stress (Bowler et al., 1991; Breusegem et al., 1999; Du et al., 2001). Taken together, these results indicated that enhanced SOD activity in plants can increase plant resistance to a variety of stressors.

Wheat (Triticum aestivum) is one of the world’s most important crops, accounting for more than half of total human food consumption (Zhang et al., 2009; Yin et al., 2018a; Yin et al., 2018b). However, its safe production is seriously threatened by natural disasters, such as drought, salinity and extreme temperature, which cause a significant decline in annual wheat yield and quality (Siddiqui et al., 2017). Correspondingly, increasing the resistance of wheat to salt stress is one of the most important objectives of the breeding program (Savadi et al., 2017). Analysis of the SOD gene in wheat can provide critical information for genetic improvement of resistance. So far, the response of the wheat SOD (TaSOD) gene family and expression of each gene under different stress conditions has not been examined on a genome-wide level. In this study, we performed genome-wide identification of the SOD gene family in wheat and comprehensively analyzed their phylogenetic relationships, distribution within the genome, gene structure arrangement, composition of motifs, expression profiles in different tissues, and their expression patterns in response to various abiotic stressors. Identification and functional analysis of the wheat SOD family provides a the foundation for further research on wheat stress resistance.

Materials and Methods

Identification of wheat SOD gene family members

A computer-based methods were used to identify members of the SOD gene family from wheat reference genome IWGSC RefSeq v1.1 (https://wheat-urgi.versailles.inra.fr/Seq-Repository/Assemblies). A total of eight Arabidopsis SODs (AtSODs), twelve maize SODs (ZmSODs), and eight rice SODs (OsSODs) protein sequences were retrieved from the Arabidopsis Information Resource (TAIR10) database (http://www.arabidopsis.org/index.jsp), the Maize Genetics and Genomics Database (MaizeGDB) (https://www.maizegdb.org/), and the Rice Genome Annotation Project (RGAP) database (http://rice.plantbiology.msu.edu/), respectively. This information was used to identify SOD genes in wheat. Two methods were utilized to search the wheat protein sequences. The first method used a Hidden Markov Model (HMM) to search against wheat protein sequences and the second method used BLASTP (E-value <1e−5) to search SOD proteins compared against the wheat genome, followed by Pfam (v31.05) (http://pfam.sanger.ac.uk/search) to test whether the obtained sequences contained a SOD specific structurally conserved domain and to determine the number of SOD gene family members.

Chromosomal locations and syntenic analyses

The wheat genome GFF3 gene annotation file was obtained from the wheat database IWGSC v1.1 and the gene annotation of wheat SODs (TaSODs) was extracted from the GFF3 file. The start and end location information of TaSODs in corresponding to chromosomes was used to produce a physical map using MapInspect software.

Proteins characterization of predicted TaSODs

Characterization analysis of TaSODs were performed by using the protein identification and analysis tools on the ExPASy server10 (https://prosite.expasy.org/) (Artimo et al., 2012). The characteristics of protein length, isoelectric point (pI), molecular weight (MW), instability index, atomic composition, and amino acid composition were predicted. The online tools TMHMM (http://www.cbs.dtu.dk/services/TMHMM/) and SignalP4.1 (http://www.cbs.dtu.dk/services/SignalP/) were used to predict transmembrane domains and signal peptides of TaSODs (Nielsen, 2017). Subcellular localization prediction of TaSODs was performed using Plant-mPLoc (http://www.csbio.sjtu.edu.cn/cgi-bin/PlantmPLoc.cgi) (Chou & Shen, 2010). TaSOD members were modeled three-dimensionally using the Phyre2 (http://www.sbg.bio.ic.ac.uk/phyre2/html/) server in the intensive mode (Kelley et al., 2015).

Phylogenetic analyses of TaSODs

Phylogenetic relationship were inferred using a Maximum Likelihood (ML) method based on the LG model with MEGA7.0 software (Kumar, Stecher & Tamura, 2016). A midpoint rooted base tree was produced using the Interactive Tree of Life (IToL, version3.2.317, http://itol.embl.de) (Fang et al., 2019).

Analysis of TaSODs motifs and gene structures

Annotation information of TaSODs was interpreted using GSDS version 2.0 (http://gsds.cbi.pku.edu.cn/index.php) to produce TaSODs gene structure, intron/exon distribution, and intron/exon boundaries (Hu et al., 2014). Conserved TaSODs gene sequences were identified using a MEME suite analysis (version 4.9.1) and MAST Primer Search (http://meme-suite.org) software tools (Bailey et al., 2015). The parameters were established using known SOD protein sequences, including AtSODs, OsSODs, and ZmSODs, and were then applied to identify conserved TaSODs as follows: each sequence comprised any number of non-overlapping occurrences of each motif, the number of different motifs was 20, and motif width ranged from 6 to 50 amino acids. The functions of these predictive motifs were analyzed using InterPro (http://www.ebi.ac.uk/interpro) and SMART (http://smart.embl-heidelberg.de/), and TBtools software (https://github.com/CJ-Chen/TBtools) was used for graphical visualization.

Multiple conditional transcriptome analysis of TaSODs

Original RNA-seq data from multiple conditional transcriptome analyses were download from the NCBI data base and mapped to the wheat reference genome which reference by hisat2. After this, genes were assembled using Cufflinks to assess expression levels of TaSODs (normalized fragments per kilobase of the exon model per million mapped reads). The R package “pheatmap” was used to produce a heatmap of expression profiles for TaSODs (Song et al., 2019).

Wheat seedling growth and stress treatments

Seeds of Emai 170 (a hexaploid common wheat variety) were surface sterilized with 1% hydrogen peroxide, thoroughly rinsed with distilled water, and germinated for 2 days in a 20 ° C incubator (Ma et al., 2016). According to the reported method, the seedlings were transferred and cultured in a continuously ventilated 1/4 strength Hoagland nutrient solution (Zhu et al., 2015). After 3 days, the Hoagland solution strength was increased to 1/2. After a further 3 days, the plants were treated with 150 mM sodium chloride (NaCl), 20% polyethylene glycol (PEG) and 180 mM mannitol. The pH of the nutrient solution was adjusted to 6.0 every 2 days by using 1 M KOH or 0.2 M H2SO4. During the treatment, the growth environment was set at 26 ° C and 16 hours/8 h (day/night). Three biological replicates were set per treatment. Leaves and roots were collected at 2, 4, 8, 12, 24, 72, 120 h after treatment with three seedlings that were biologically mixed together. Each treatment consisted of three replicates, each of which included three technical replicates. Next, the sample was immediately frozen with liquid nitrogen and stored at −80 °C.

Real-time quantitative PCR and data analysis

To elucidate the developmental and tissue-specific expression profiles of the SOD gene in wheat, quantitative real-time PCR (qRT-PCR) was performed to detect the expression level of the SOD gene. Total RNAs from different tissues (leaf and root) and stress-treated leaves were reverse transcribed with 5X All-In-One RT Master Mix (Perfect Real Time) kit (ABM, Canada) into cDNAs for qPCR analysis. Gene-specific primers were designed using Primer Premier 5.0. For qRT-PCR assay, the cDNA was diluted to 400 ng/µL with ddH2O. The reaction system contained 5 µL of 2 ×SYBR green Mix, 0.5 µL of each primer (10 µM), 0.5 µL of template (about 400 ng/µL) and 3.5 µL of ddH2O to make a total volume of 10 µL. The protocol was carried out as follows: pre-denaturation at 94 °C for 3 min (step 1), denaturation at 94 °C for 10 s (step 2), primer annealing/extension and collection of fluorescence signal at 60 °C for 30 s (step 3). The next 40 cycles start at step 2. Three biological replicates were performed for each sample, with three technical replicates repeated each. Relative expression levels were determined using the 2−ΔΔCt method (Livak & Schmittgen, 2001). The expression level of TaSOD genes were plotted using Origin software.

Results

Identification of SODs from the wheat genome

In order to identify wheat SOD proteins (TaSODs), twenty-eight known SOD proteins, including eight AtSODs, twelve ZmSODs and eight OsSODs (Kliebenstein, Monde & Last, 1998; Dehury et al., 2013; Krishna et al., 2014), were collected as query sequences to a conduct BLASTP searches against the wheat reference genome IWGSC v1.1 (E-value < 1e–5). Candidate hits were further confirmed using Pfam and local BLASTP searches with the core motif (E-value < 1e–5) to further confirm whether the TaSODs contained the SOD domain. A rigorous bioinformatic screening identified 26 reliable TaSODs (Table 1), including eleven, five, and ten loci from the sub-genomies of A, B, and D, respectively. At this point, fifty-four SODs proteins were obtained from the four plant genomes (Arabidopsis, rice, maize, and wheat), as shown in detail in the supplemental information (Table S1). The sequences were re-named in ascending order based on their phylogenetic relationships of the OsSODs (Liu et al., 2018). The analysis of 26 wheat SOD genes produced 17 Cu/Zn-SODs (TaSOD1.1a-TaSOD1.11b), 6 Fe-SODs (TaSOD2.4-TaSOD2.9), and 3Mn-SODs (TaSOD2.1-TaSOD2.3). This result was consistent with the protein annotation information. Furthermore, alternative splicing isoforms were observed in TaSOD1.1, TaSOD1.5, TaSOD1.6, TaSOD1.7, TaSOD1.8 and TaSOD1.11.

Gene structure and chromosomal distribution of wheat genes encoding SOD proteins

In order to investigate the gene structure of TaSODs, we analyzed their GFF3-formatted annotation and observed that all TaSODs had introns. A sequence alignment of 26 TaSODs using DNAMAN software revealed low homology between the 26 proteins, and the highly conserved region was mainly concentrated at the C-terminus, which may be the key region for the function of TaSODs (Fig. 1). Exon-intron structural diversity frequently plays a key role in the evolution of gene families and can provide additional evidence to support phylogenetic grouping (Qu & Zhu, 2006; Liu, White & Macrae, 2010). The exon-intron structure of the TaSOD genes was further examined based on its evolutionary classification. As shown in Fig. 2B, all TaSOD genes contained introns, and number of introns ranged from four to sever. Sever TaSOD members (TaSOD1.9, TaSOD1.10, TaSOD1.11a, TaSOD1.11b, TaSOD2.5, TaSOD2.6, and TaSOD2.8) contained the largest number of introns (sever introns), while the smallest number (four introns) was observed only in TaSOD1.5b. As expected, the exon/intron distribution patterns were similar among SOD members within each clade of the phylogenetic tree. For example, the TaSOD2.1, TaSOD2.2, and TaSOD2.3 had the same number of exons/introns splicins and similar length.

Table 1 Predicted sequence features of TaSODs.

Group	Designation	Gene ID	aLength	bMW	cpI	dIns.	eAli.	fGRAVY	gSub.	
SOD1	TaSOD1.1a	TraesCS2A02G121200.1	157	15.70149	5.81	17.3	82.55	−0.015	hCyt.	
TaSOD1.1b	TraesCS2A02G121200.2	141	14.1218	6.01	20.93	81.56	−0.003	Cyt.	
TaSOD1.2	TraesCS2A02G399000.1	311	32.3006	5.39	38.55	86.05	−0.001	Cyt.	
TaSOD1.3	TraesCS2B02G417000.1	308	32.15458	5.66	40.23	87.82	0.028	Cyt.	
TaSOD1.4	TraesCS2D02G123300.1	152	15.09177	5.7	17.84	80.79	−0.028	Cyt.	
TaSOD1.5a	TraesCS2D02G396500.1	309	32.16249	5.39	39.71	86.6	0.01	Cyt.	
TaSOD1.5b	TraesCS2D02G396500.2	301	31.3796	5.57	39.85	85.98	0.002	Cyt.	
TaSOD1.6a	TraesCS4A02G065800.1	164	16.57647	6.58	24.26	83.29	−0.175	Cyt.	
TaSOD1.6b	TraesCS4A02G065800.2	212	22.20282	8.81	26.24	78.21	−0.302	Cyt.	
TaSOD1.7a	TraesCS4B02G243200.1	164	16.68561	6.39	23.82	80.91	−0.171	Cyt.	
TaSOD1.7b	TraesCS4B02G243200.2	174	18.04719	7.23	23.56	76.26	−0.271	Cyt.	
TaSOD1.8a	TraesCS4D02G242800.1	146	15.1378	5.93	24.82	83.49	−0.2	Cyt.	
TaSOD1.8b	TraesCS4D02G242800.2	164	16.6626	6.39	24.91	85.67	−0.112	Cyt.	
TaSOD1.9	TraesCS7A02G292100.1	201	20352.9	5.22	24.45	93.23	0.132	Cyt.	
TaSOD1.10	TraesCS7B02G197300.1	201	20.32292	5.35	22.8	94.18	0.156	Cyt.	
TaSOD1.11a	TraesCS7D02G290700.1	201	20.25075	5.35	23.96	93.23	0.13	Cyt.	
TaSOD1.11b	TraesCS7D02G290700.2	202	20.32183	5.35	23.89	93.27	0.139	Cyt.	
SOD2	TaSOD2.1	TraesCS2A02G537100.1	231	25.29893	7.89	29.8	91.73	−0.245	iMit.	
TaSOD2.2	TraesCS2B02G567600.1	225	24.60303	7.14	29.35	90.71	−0.278	Mit.	
TaSOD2.3	TraesCS2D02G538300.1	231	25.27483	7.91	31.71	90.48	−0.282	Mit.	
TaSOD2.4	TraesCS4A02G390300.1	261	29.81302	7.23	59.33	82.22	−0.429	jChl.	
TaSOD2.5	TraesCS4A02G434000.1	390	42.91936	9.41	50.74	71.92	−0.526	Chl.	
TaSOD2.6	TraesCS7A02G048600.1	392	43.40094	9.31	54.79	70.56	−0.544	Chl.	
TaSOD2.7	TraesCS7A02G090400.1	260	29.798	6.84	57.55	82.54	−0.427	Chl.	
TaSOD2.8	TraesCS7D02G043000.1	391	43.32193	9.17	55.37	68.98	−0.547	Chl.	
TaSOD2.9	TraesCS7D02G086400.1	260	29.83994	6.87	58.86	82.88	−0.432	Chl.	
Notes.

a Length (Amino acid length).

b MW (Molecular weight, KD).

c pl (Isoelectric point).

d Ins. (Instability index).

e Ali. (Aliphatic index).

f GRAVY (Grand average of hydropathy).

g Sub. (Subcellular localization).

h Cyt. (Cytoplasm).

i Mit. (Mitochondria).

j Chl. (Chloroplast).

Figure 1 Multiple alignment of TaSOD proteins of functional domain.

(A) TaSOD1 (Cu/ZnSODs) subfamily sequence alignment. The motif1 conserved domain is marked in the figure. (B) TaSOD2 (Fe-SODs and Mn-SODs) subfamily sequence alignment. The motifs of motif4 and motif6 are marked in the figure. And metal-binding domain are also labeled.

Figure 2 Phylogenetic analysis, gene structure, and conserved motifs of TaSODs.

(A) The phylogenetic tree of all SOD genes in Triticum aestivum. The tree was created with bootstrap of 1000 by maximum likelihood (ML) method in MEGA7. (B) The exon-intron structure of SOD genes in Triticum aestivum. Exon-intron structure analyses were conducted using the GSDS database. Lengths of exons and introns of each TaSOD gene are displayed proportionally (Figure S1). (C) The motif compositions of TaSODs were identified by MEME. Model exhibition of motif compositions in SOD amino acid sequences using MAST. Each motif is indicated with a specific color. symbol represents the Cu/Zn-SOD domain, ⋆ symbol represents the Fe_N domain, □ symbol represents the Fe_C domain.

Information regarding to TaSODs were extracted from the GFF3 reference file of the wheat genome to determine the chromosomal location of the TaSOD genes. Based on this extracted physical location (Supplemental information: Table S3), a chromosomal map of TaSOD genes was produced using the MapInspect software. The SOD gene map of the wheat genome showed SODs only on chromosomes 2, 4, and 7. The density of these loci was highest on chromosome 2 was higher with 38.46% of all SOD genes (Fig. 3).

Figure 3 Chromosomal localization of the 26 TaSODs genome.

Different classes of TaSODs are represented in different colors. Red represents TaSOD1 and blue represents TaSOD2.

TaSODs protein features

The amino acid sequences of 26 TaSODs proteins were submitted to the ExPASy server10 (http://www.expasy.org/tools/) online analytical system for analysis of biochemical characteristics such as isoelectric point (pI), relative molecular mass (MW) and instability index (Table 1). The results showed that the TaSODs had an average theoretical pI of 6.69 with a from 5.22 to 9.42. Protein length ranging from 141 to 392 amino acids with an average of 236 amino acids and an average molecular weight of 25.14396 kDa (range from 14.1218 kDa to 43.40094 kDa). In line with previous results, all Cu/Zn-SODs were acidic whereas Fe-SODs and Mn-SODs were basic or alkaline (Dehury et al., 2013; Zhang et al., 2016a). In the present study, most of the SOD1 were acidic in character, apart from two SOD1 enzymes (SOD1.6b and SOD1.7b). Most of the SOD2 enzymes were alkaline proteins except for two SOD2 enzymes(SOD2.7 and SOD2.9). The GRAVY analysis showed that the SOD2 subfamily contained only hydrophilic proteins, whereas the SOD1 subfamily comprised hydrophilic and six SOD1 (TaSOD1.3, TaSOD1.5a, TaSOD1.9, TaSOD1.10, TaSOD1.11a, and TaSOD1.11b) hydrophobin proteins. Interestingly, all Cu/Zn-SODs of wheat that were predicted to be localized in the cytoplasm were classified as acidic amino acids. All Fe-SODs were predicted to be localized in the chloroplasts and comprised mostly alkaline amino acids, whereas all Mn-SODs were composed with alkaline amino acids and located in mitochondria and were composed of alkaline amino acids.

Phylogenetic relationship analyses

To gain a better understanding of the evolutionary history and evolutionary relationships of the SOD gene family in wheat, a phylogenetic tree was generated using a ML method with the full-length amino acid sequences. The phylogenetic tree revealed that these SOD proteins could be classified into two major groups: SOD1 and SOD2. Moreover, we found that the SOD1 subfamily consisted of Cu/Zn-SODs and the SOD2 subfamily consisted of Fe-SODs and Mn-SODs. Based on the phylogenetic tree, we observed that SOD proteins within the same subfamily were clustered together, whereas Fe-SODs and Mn-SODs were divided into one sub-groups (Fig. 4), implying that Fe-SODs and Mn-SODs originated from a common ancestor (Alscher, Erturk & Heath, 2002). The SOD1 group consisted of 17 TaSODs (TaSOD1.1a to TaSOD1.11b), 3 from AtSODs, 6 from ZmSODs, and 5 from OsSODs. Similarly, the SOD2 proteins included 9 TaSODs (TaSOD2.1 to TaSOD2.9), 5 AtSODs, 6 ZmSODs, and 3 OsSODs. Moreover, we also could find that the dicot SODs (Arabidopsis) have more closely phylogenetic relationship related to monocot SODs (wheat, rice and maize) in each clade when compared with all plants.

Figure 4 Phylogenetic relationship of TaSODs, OsSODs, AtSODs, and ZmSODs.

Protein sequences were aligned using ClustalW2 sequence alignment program and the phylogenetic tree was constructed by software MEGA7 used to create maximum likelihood (ML) under the LG model. The tree was constructed with 1,000 bootstrap replications. Different groups were marked by different colors, and the SOD from wheat, rice, maize and Arabidopsis were distinguished with different color and shape.

Conserved motifs and clustering analyses of TaSODs

To investigate the evolutionary relationship of SODs in wheat, a phylogenetic tree was constructed by aligning the 26 TaSODs. These TaSODs clustered in two groups (TaSOD1 and TaSOD2), which was highly consistent with the type of their metal cofactors (Fig. 2A). To further examine structural diversity and predict the function of the TaSOD proteins, 20 motifs in TaSODs were identified using MEME software and were further annotated using Inter ProScan 5 software (Fig. 2C). Details of these 20 motifs are shown in the supplemental information: Table S2. Previous studies reported that the SOD gene family typically contains highly conserved domains involved in metal binding (Perry et al., 2010). The motifs 1, 2, 3, 6, 10, 11, and 17 together constitute the SOD conserved sites. The motifs 1, 2, 10, and 13 are Cu/Zn-SODs conserved domains, and the motifs 3, 6, 11, and 17 are conserved domains of Fe-SODs and Mn-SODs.

The same subfamily-associated pattern was observed regarding common motifs. All TaSODs in the SOD1 subfamily contained motifs 1 and 5, however, motif 5 is not a conserved domain of TaSOD. All members of SOD2 subfamily contained motifs 3 and 6. In addition, the Cu/Zn-SODs conserved domains (motif 1) were analyzed to understand the relationship between TaSOD1 and SODs in other species. An alignment of all Cu/Zn-SODs conserved domains of 17 TaSOD1 was termed a conserved motif 1. This result showed eight conserved amino acids (glycine, leucine, histidine, aspartic acid, serine, threonine, asparagine, and proline) in the Cu/Zn-SODs conserved domains motif. The conserved motif of Fe-SODs and Mn-SODs conserved domains site were termed as motif 3 and motif 6, respectively. Motif 3 had eight conserved amino acids (valine, proline, tyrosine, alanine, leucine, glutamic acid, serine, and histidine), and motif 6 included the conserved metal-binding domain “DVWEHAYY” of the Mn-SODs and Fe-SODs. Sequences, locations, and logos of the conserved motifs (motif 1, motif 3, and motif 6) in the TaSOD proteins are shown in Fig. 5. The data analyses supported our results. All of the identified wheat genes contained conserved domains of the sod family. In line with previous studies on different plant species, the SOD gene family in wheat contained characteristic amino acids, including a series of highly conserved active site residues that play roles in the sequence-specific binding of mental ions.

Figure 5 Conserved motifs of TaSODs.

The number on x axis indicates the position of amino acid, and the number on Y axis indicates represents the conservation of the protein. The height of a letter indicates its relative frequency at the given position (x-axis) in the motif.

Functional categorization of wheat superoxide dismutase genes functional classification by GO annotation

Since superoxide dismutase function could scavenge intracellular superoxide anion radicals to protect cells from damage by reactive oxygen species (ROS) (You & Chan, 2015). Gene ontology (GO) annotation was performed to predict TaSOD functions (Gotz et al., 2008; Yin et al., 2018a; Yin et al., 2018b). Enrichment analysis showed that most TaSODs were annotated under GO terms ‘superoxide dismutase activity’ (GO:0004784), ‘oxidation–reduction process’(GO:0055114), and ‘removal of superoxide radicals’ (GO:0019430). Several TaSOD-I (e.g., TaSOD1.1 and TaSOD1.4) were annotated under ‘gluconeogenesis’ (GO:0006094), ‘glycolytic process’ (GO:0006096), and ‘response to cadmium ion’ (GO:0046686). In TaSOD-II subfamily, eg. TaSOD2.1 and TaSOD2.3, were annotated under ‘response to salt stress’ (GO:0009651), ‘response to zinc ion’(GO:0010043), and ‘defense response to bacterium’ (GO:0042742). GO enrichment analysis showed that multiple functions of TaSODs (Fig. 6, Supplemental information: Table S4). These results are in accordance with those of previous studies, which suggested that plant antioxidant is a complex regulating network that involves in substance and energy metabolism, and defense process (Tewari et al., 2008; Karuppanapandian et al., 2011).

Figure 6 Figure Functional GO annotation analysis of TaSOD genes.

Multiple conditional transcriptome analysis of TaSODs

We performed comprehensive microarray analyses to estimate the expression level of each TaSOD gene in different organs. RNA-seq data (Supplemental information: Table S5) from multiple conditional transcriptome analysis were downloaded from the NCBI database and mapped to the wheat reference genome using hisat2. After this, genes were assembled using Cufflinks software to assess the expression levels of TaSODs. The R package “pheatmap” was used to produce a heatmap of the wheat SOD genes. Previous studies showed that different pathways of SOD enzyme expression regulation patterns are unique and interact with each other (Dou et al., 2010). As shown in Fig. 7, the SOD gene family members were expressed in different tissues, and expression patterns differed between the SOD gene family members. The expression patterns were similar within subfamilies. The TaSODs can be classified into two groups: one group contains members that are widely expressed in numerous tissues, at different developmental stages, and under different treatment conditions, and the other group contains members that are not consistently expressed or highly induced only under certain conditions. Interestingly, we also found that most Fe-SODs were not highly expressed in various tissues and under different environmental abiotic stresses. Furthermore, in the salt stress environment, the expression levels of most Cu/Zn-SODs and Mn-SODs under salt stress conditions. In contrast, we clearly found that Cu/Zn-SODs and Mn-SODs were significantly up-regulated under drought and high temperature conditions. In particular, TaSOD1.1a and TaSOD1.4 showed the highest expression levels under drought and heat stress.

Figure 7 Multi-conditional transcriptome analysis of TaSODs.

Expression level of wheat SOD genes in different tissues under different abiotic stress environments. The R package ‘pheatmap’ was used to generate heat maps based on the log2(FPKM+1) values. The depth of the color in the figure reflects the strength of gene expression.

Figure 8 Real-time quantitative reverse transcription-polymerase chain reaction (qRT-PCR) analyses of TaSOD genes in plants under abiotic stresses NaCl, PEG and mannitol treatment in both leaf and roots for the indicated time periods.

Time periods are shown on the x-axis and the expression levels on the y-axis. Different tissues are displayed in different color boxes, with red squares representing leaf tissue and black squares representing root tissue. The data were analyzed by three independent repeats, and standard deviations were shown with error bars. The expression level of TaSOD genes were plotted using Origin software.

Expression analysis of wheat SOD genes in response to salt and drought

High salinity and drought are the major environmental stresses that frequently affect the growth and development of plants under various natural conditions (Xia et al., 2012). To simulate salt stress and drought conditions, we treated wheat seedlings with 0.15 M/L NaCl, 0.18 M/L mannitol, and 20% PEG at one-leaf stage. After treatment, the seedling growth status changed considerably. At 12 h, the leaves softened and turned yellow, and growth inhibition started. After 72 h, leaf moisture was lost, and the leaves became brittle (Fig. S2).

To understand how the TaSOD genes are involved in salt and drought stress responses, qRT-PCR was used to analyze the expression profiles of these genes under different conditions for different time in the leaves and roots. Our data show that under adverse conditions, there is a complex regulatory mechanism for the expression of TaSOD genes (Fig. 8). In leaves, the expressions of 8 genes under three treatments were variations, whereas most genes expressions were downregulated during early stage of PEG treatment. The expression of TaSOD1.7 was dynamic, increasing before 8 h, then gradually decreasing, and finally dropping to the lowest point. In root tissue, obviously, most TaSOD genes were up-regulated in response to mannitol stress. Among them, five members (TaSOD1.7, TaSOD1.9, TaSOD1.11a, TaSOD2.1 and TaSOD2.3) exhibited more than 2- to 5- fold decreases. During salt treatment, the expression levels of most TaSOD genes exhibited slight variations, whether leaf or root tissue. However, TaSOD1.7 gene showing obvious differential expression in response to NaCl stress (TaSOD1.7 was up-regulated in leaf, while it was down-regulated in root; Fig. 8).

Homology modeling of TaSODs

All 26 wheat TaSOD members were three-dimensionally modeled using the Phyre2 server in intensive mode (Fig. 9). Predicted models were based on following templates to heuristically maximize the alignment coverage, percentage identity, and confidence score for the tested sequences: template c2q2IB were used in TaSOD1.1a TaSOD1.1b and TaSOD1.4 modeling, template c1jkqD in TaSOD1.2, TaSOD1.3, TaSOD1.5a and TaSOD1.5b modeling, template d2c9val in TaSOD1.6a, TaSOD1.6b, TaSOD1.7a, TaSOD1.7b, TaSOD1.8b, TaSOD1.9 and TaSOD1.10 modeling; template d1srda in TaSOD1.11a and TaSOD1.11b models .And template c4c7uB in TaSOD2.1, TaSOD2.2, and TaSOD2.3 modeling; template c6bejA in TaSOD2.4 ,TaSOD2.5, TaSOD2.6, TaSOD2.7, and TaSOD2.9 modeling; template c1xreB in TaSOD2.8 modeling. The quality of models was validated using a Ramachandran plot analysis in which 80% of residues were within the permitted area, indicating fairly good structures of the models. However, it was apparent that in order to construct more reliable, and realistic models, more experimentally solved structures are required from SOD family proteins, particularly from plant SODs.

Figure 9 Predicted 3D models of TaSOD proteins. Models were generated by using Phyre2 server at intensive mode.

Models were visualized by rainbow color from N to C terminus and organized in order as gene sequence (TaSOD1.1 to TaSOD2.9). Different sub-family proteins have similar protein models.

In the SOD1 subfamily, the secondary structures of modeled wheat proteins were primarily β-strands (26–41%) whereas α-helices occurred at only 3–14%. However, in the SOD2 subfamily, constituted the secondary structures of modeled wheat proteins were primarily α-helices primarily (45–60%), whereas β-strands occurred at only 8–12%. This is in line with the results reported in previous studies (Keerthana & Kolandaivel, 2015). Moreover, to assess similarity or divergence of generated models, structures were superimposed to calculate the percentages of structure coverage. The superimposed SOD1 subfamily models showed 69–100% structural coverage and the superimposed SOD2 subfamily models showed 51–89% structural coverage. In the SOD1 subfamily, we found that structural coverage of TaSOD1.1b and TaSOD1.4 was 100%. However, in the SOD2 subfamily, some models such as TaSOD2.5 (51%), and TaSOD2.6 (51%) showed low structural similarity but were above the twilight zone (30%). Taken together, our results suggest that SODs from each genome donor may either have been ancestrally similar to each other or originally divergent SODs could have been stabilized over a long domestication process resulting in changes in protein structures and functionality.

Discussion

Wheat is the second largest food crop in the world and is of great significance to human life (Ling et al., 2018). However, wheat yield is affected by various adverse environments. Superoxide dismutase (SOD) play important roles in multiple processes of plant growth and resistance against environmental stressors (Fernández et al., 2011). Interestingly, only a small fraction of SOD genes have been identified in plants. Genome-wide analysis is an important approach for elucidating the biological roles of the SOD gene family members in a given plant species. The SOD gene family has been reported to be widely distributed in different plant species, such as Arabidopsis (Kliebenstein, Monde & Last, 1998), longan (Lin & Lai, 2013), rice (Dehury et al., 2013; Krishna et al., 2014), poplar (Molina et al., 2013), banana (Feng et al., 2015), pear (Wang et al., 2018), tomato (Feng et al., 2015), cotton (Zhang et al., 2016a), and cucumber (Zhou et al., 2017). Interestingly, however, there is no comprehensive analysis of the SOD gene family in wheat (Triticum aestivum). The availability of whole genome sequence of the species aided in genome-wide characterization of the SOD genes, which may further be used to improve the crop yield on field.

The basic pipeline for identifying SOD genes including Blast search the known proteins of related families and pfam search, was followed as reported earlier for other plants (Zhou et al., 2017; Deepika, Neha & Kashmir, 2019). In the present study, a total of 26 SODs genes were identified in wheat, which cover the three major types of plant SOD genes, including 17 Cu/Zn-SODs, 6 Fe-SODs, and three Mn-SODs (Table 1). The number of SOD genes varies between plants, and previous studies revealed that the numbers of SOD genes in Arabidopsis, rice, sorghum, and tomato had eight (3 Cu/Zn-SODs, 2 Mn-SODs, and three Fe-SODs), eight (5 Cu/Zn-SODs, one Mn-SOD,and 2 Fe-SODs), eight (5 Cu/Zn-SODs, one Mn-SOD, and two Fe-SODs), and nine SOD gene (four Cu/Zn-SODs, one Mn-SOD, and four Fe-SODs), respectively. There are large differences in the genome size, and the number of SOD genes varies among these plant species; this variation in the number of SOD genes, however, does not correspond to the variation in genome size. Differences in the number of SOD genes between plant species may be attributed to gene duplication, which comprises tandem and segmental duplications and plays a crucial role in the expansion of SOD genes for diversification. Gene duplication of SOD genes was also found in different plant species (Zhang et al., 2016a; Wang et al., 2016b; Wang et al., 2016b). Therefore, these results imply that TaSOD duplication events play key role in gene evolution.

Gene structure analysis revealed four to seven introns in the 26 wheat SOD genes (Fig. 2B). A previous study showed that plant SOD genes had highly conserved intron patterns, and most cytosolic and chloroplast SODs harbored seven introns (Fink & Scandalios, 2002). In our study, seven members (TaSOD1.9, TaSOD1.10, TaSOD1.11a, TaSOD1.11b, TaSOD2.5, TaSOD2.6, and TaSOD2.8) of the SOD gene family were predicted to contain seven introns (Fig. 2B). The divergence of TaSOD gene structures may be due to the mechanisms including exon/intron gain/loss, exonization/pseudoexonization, and insertion/deletion as suggested in a previous study (Xu et al., 2012); moreover, the SOD members in each clade of the phylogenetic tree displayed similar exon-intron organization patterns (such as TaSOD1.6a and TaSOD1.8b; TaSOD2.1 and TaSOD2.3), suggesting that they may have similar functions related to various abiotic stressors.

Phylogenetic analysis revealed a close relatedness between Cu/Zn-SODs and Fe-SODs/Mn-SODs members. Comparative phylogenetic analyses of SOD proteins of wheat and three other plant species (Arabidopsis, maize, and rice) showed that both formed two separate groups based on the bootstrap values, which is consistent with the results of previous studies (Wang et al., 2016b; Liu et al., 2018). Most of the results regarding subcellular localization of SODs confirmed the phylogenetic results. All Cu/Zn-SODs were grouped in the subfamily SOD1 and were predicted to be located in the cytoplasm. Chloroplast Fe-SODs and mitochondrial Mn-SODs clustered in sub-group 2. In addition, phylogenetic analysis of other species of SODs found that for most of the wheat SODs homologous sequences can be found in Arabidopsis, maize, or rice (Fig. 4), suggesting that TaSODs probably have the same functions as SODs in other plant species. Interestingly, related SOD orthologous genes from different plants clustered together, but monocots and dicotyledons showed distinct separation. The unique evolutionary relationship of monocotyledonous SOD and dicotyledon can explain the common ancestry of two groups of plants on this basis (McClung, 2010).

Abiotic factors like drought, heat, cold, salinity pose a serious threat to the crop yield of wheat crops, thus expression analysis of SOD genes under drought and heat stress were studied. Previous reports on A. thaliana, Gossypium raimondii, and Cucumis sativus suggest the roles that SODs play in overcoming the stress (Kliebenstein, Monde & Last, 1998; Feng et al., 2016; Zhou et al., 2017; Zhang et al., 2016b). Transcriptome analyses of SOD family genes revealed that various environmental stressors had a regulatory effects on the expression of TaSOD genes. Different TaSOD genes were differentially expressed in response to the same environmental stressors, and there were also differences in the expression regulation of the same gene under different stressors. This also suggests that different TaSOD proteins may exhibit different mechanisms of action in response to adverse effects (Bolwell, 1998; Bubliy & Loeschcke, 2005). In our study, qRT-PCR was used to analyze wheat expression levels under drought and salt stress conditions to understand their involvement in stress response. When the plants were exposed to longer duration of drought stress, four genes (TaSOD1.1a, TaSOD1.4, TaSOD2.1, and TaSOD2.3) showed up-regulation in the expression levels in leaf and five genes (TaSOD1.7, TaSOD1.9, TaSOD1.11a, TaSOD2.1, and TaSOD2.3) showed up-regulation in the expression levels in root. On exposure to salt stress, TaSOD1.7 gene showed significant increase in the expression levels in leaf, whereas the other genes showed decrease in the expression but that change was not significant. The expression study also explained the role of SODs in overcoming abiotic stress in wheat species.

Conclusions

In this study we have analyzed the wheat genome to identify and characterize the SOD gene family by using a broad range of bioinformatic tools. SOD are the core of antioxidant enzymes and among the first to participate in the scavenging of reactive oxygen species. It is widely involved in the response of plants to stress. It are widely involved in the response to plants to stress. The results of this study increase our understanding of the evolutionary relationships in the SOD family and also serve as the basis for the functional identification of wheat SOD proteins. In conclusion, our works have provided comprehensive information about the 26 SOD genes family in wheat, including gene structures, chromosome localization, phylogenetic relationships, and expression profiling of these gene families indicated that TaSOD genes are involved in the regulation of plant tissue development and likely have important role in response to abiotic stress. This systematic genome-wide identification provides basis for future studies on the function of TaSOD proteins in biological processes, and providing a potential basis for wheat in drought and salt stress breeding improvement.

Supplemental Information

Table S1 SOD genes found in Arabidopsis thaliana, Oryza sativa, and Zea mays

Click here for additional data file.

Table S2 Annotation of putative of TaSODs identified by MEME

Click here for additional data file.

Table S3 Location TaSODs genes on Chinese Spring

Click here for additional data file.

Table S4 Functional annotation of wheat SOD genes

Click here for additional data file.

Table S5 The FPKM data of TaSOD genes in different tissues and environment

Click here for additional data file.

File S1 The gene sequences used in this research

Click here for additional data file.

Figure S1 The exon/intron organization of TaSOD

Click here for additional data file.

Figure S2 Wheat phenotype after drought and salt stress treatment

Click here for additional data file.

We thank Prof. Yongli Qiao for comments on the initial project design and data analysis.

Additional Information and Declarations

Competing Interests

Author Contributions

Data Availability

The authors declare there are no competing interests.

Wenqiang Jiang conceived and designed the experiments, performed the experiments, analyzed the data, prepared figures and/or tables, authored or reviewed drafts of the paper, approved the final draft.

Lei Yang, Yiqin He, Wei Li and Huaigu Chen performed the experiments, analyzed the data, prepared figures and/or tables, authored or reviewed drafts of the paper, approved the final draft.

Haotian Zhang performed the experiments, analyzed the data, prepared figures and/or tables, authored or reviewed drafts of the paper, approved the final draft, comparative experimental data.

Dongfang Ma and Junliang Yin conceived and designed the experiments, performed the experiments, analyzed the data, contributed reagents/materials/analysis tools, prepared figures and/or tables, authored or reviewed drafts of the paper, approved the final draft.

The following information was supplied regarding data availability:

Transcriptome data is available in Table S4.

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
