# Peer review of "Genome-wide identification and transcriptional expression analysis of superoxide dismutase (SOD) family in wheat (Triticum aestivum)"

_PeerJ, doi:10.7717/peerj.8062_

## Round 0.1 · original submission · Major Revisions

· Academic Editor

Major Revisions

The manuscript is being returned for major revision at this point because there are too many grammatical errors in the presentation, requiring a reviewer to decipher the clear meaning intended and to additionally interpret the science trying to be presented. I have included some mark-up from my initial reading of the manuscript; however, I recommend that a language service review this manuscript to create an easier read for the benefit of reviewers. It is for your benefit that reviewers be able to focus on the science of the manuscript rather than the language; therefore, it is being returned to await an improved version. In returning this manuscript I must stress that the scientific content has yet to be evaluated and will be forwarded to qualified reviewers once it is returned. Structurally, it is best to provide tables and figures as separate attachments rather than having them included within the body of the manuscript, similar to what is done with supplemental data files. I look forward to your revised manuscript.

---

## Round 0.2 · Major Revisions

· Academic Editor

Major Revisions

This manuscript still requires major revision primarily due to the language structure. I strongly recommend that an English-language proofreading be done before re-submitting. Likewise, embedding the figures and tables within the text portion of the manuscript creates difficulty in reviewing the document; usually the tables and figures are attached separately at the end of the document. Again, I recommend that a language service review this manuscript to create an easier read for the benefit of reviewers. For the reviewers that did try to understand what was presented some key observations were highlighted; please attend to their suggestions. I would strongly suggest that some additional research be incorporated into this manuscript; as stated in the abstract you are merely doing some preliminary bioinformatic background studies without presenting a stated case for using the work. For instance if this study were to be used to improve wheat, what qualities would you be looking for since an array of SODs are present. Is there a quality associated with a certain balance of the different gene families? How does this information enhance the annotation already provided in the released GFF file for provided sequences of the reference genome. Were similar sequences sought in some of the other wheat genomes of other cultivated wheat sequences made available? Wouldn’t this enhance your studies? Some of the mark-up that I did on your revised version did not change much from the previous version, so I would recommend some careful editing. Please respect the comments provided and try to Improve your future revision. I regret that this still appears to require major revision; I hope to see a much improved revision.

·

Basic reporting

Pay attention to grammatical errors.

Experimental design

no comment

Validity of the findings

no comment

Additional comments

The manuscript entitle "Genome-wide identification and transcriptional expression analysis of superoxide dismutase (SOD) family in wheat (Triticum aestivum)" identified 26 SOD genes in wheat. It is a good topic and be of siganificance for the wheat SOD protein. However, this study has the following points need to be major revision.

1. Though it is well-known that the role of SOD family about stress tolerance. I hereby recommend this study should add some newestly researches in INTRODUCTION section.
2. Pay attention to grammatical errors.
3. Line 22: Arabidopsis should be Italic.
4. Lines 25: This sentence seem to confused; please rephrase.
5. Introduction: The author did not provide any logical reason why it is important to study SOD family in Wheat. They talked about SOD and stress in the three paragraphs but didn't provide any link between that and Wheat. How can this study benefit Wheat (breeding or physiological studies).
6. Lines 179: Are there some evidence to prove C-terminus is key region for the function of TaSODs?
7. Lines 280: motif3 and motif6 should be omit.
8. Are there conserved functions for SOD genes in plants?
9. The discussion part did not discuss or properly address the findings

Therefore I whole heartedly suggest this manuscript for publication in peerj after a moderate revision.

Reviewer 2 ·

Basic reporting

The manuscript is a little colloquial, especially in the Abstract and Introduction , there is some obvious mistake,For example:
P5 L20,"Superoxide dismutases (SODs) are a family of key antioxidant enzyme enzymes", "enzyme" repetitively appeared.
P6 L38, the word "disease" should be replaced by "pathogens".
P6 L51-52, the statement about SOD lack a proper reference.
P7 L60-61, the sentence "SOD constitutes the first line of defense in ROS elimination in plant. This enzyme is ubiquitous in the plant kingdom and occurs in numerous forms." should be delated, for the similar information has been presented.
P4 L73-74, please add the Latin name of the species which firstly appear in your manuscript.
The Abstract was a flat statement, and the author should present a scientific direction based on the results of this manuscript, which can arouse their own and audiences` interest.
The manuscript should be rewrite carefully by a professional editor.

Experimental design

The RNA-seq results MUST need quantative-PCR support, it is a critical experiment for this manuscript, especially the expression pattern of SOD.
It is encouraged that the author introduce biochemistry tests of SOD enzyme activity, which they regard as important.

Validity of the findings

I agree with the authors that SOD family play an indispensable role in response to biotic and abiotic stress. Bioinformation excavation is a smart start, and optimize the released nucleotide sequence is approbatory.
The bioinformatic analysis is well organized and quite clear, and I like the Figures presented in the manuscript. (However,in Figure 6 and Figure 7, there was only title and limited legend, and I think the audience need more information from a detailed legend).

Additional comments

Based on the suggestion above, I wish the author could distinguish the kind of TaSODs that function prominently in biotic or abiotic stress. It will be valuable for guiding the wheat breeding.

Annotated reviews are not available for download in order to protect the identity of reviewers who chose to remain anonymous.

---

## Round 0.3 · Minor Revisions

· Academic Editor

Minor Revisions

The manuscript now reads well and is getting closer to an acceptable form. Because there are 26 genes described and the expression under different tissues and conditions analyzed, it may be important to extend the annotations a bit using terminology that describes the metabolic, molecular, and cellular states of the genes. I should have mentioned this in earlier correspondence; however, the language issues were too great in earlier version to see what was to be the final outcome.

The use of gene ontology annotations would go far to explain the concepts being developed for the gene family as 26 new genes were annotated and as they were followed for expression in the different tissue types. GO annotations are designed specifically for these situations to highlight the molecular function, biological process, and cellular component details of expressed sequences.

Journal manuscripts are often scanned by text-mining software that locates and extracts core data elements, like gene function. Adding standard ontology terms, such as the Gene Ontology (GO, geneontology.org) or others from the OBO foundry (obofoundry.org) can enhance the recognition of your contribution and description. This will also make human curation of literature easier and more accurate. None of this was visible.

I will set the manuscript at the ‘minor revision’ level until the annotation aspects can be addressed. Please consider this request and we look forward to the enhanced version. Thank you for your contribution.

---

## Round 0.4 · Minor Revisions

· Academic Editor

Minor Revisions

Thank you for the edits that were applied in your most recent revision of the manuscript. The manuscript reads well and is considered acceptable to move forward. However, it was noted upon further review that the wheat genome already has annotation based on the IWGSC efforts. While this paper clearly improves on that annotation for the SOD genes, acknowledgement that there is an existing functional annotation already available and a discussion would be required to highlight how the results presented compare to those already made. For example searching for "superoxide dismutase" at ensemble plants returns 20 genes ( http://plants.ensembl.org/Triticum_aestivum/Search/Results?species=Triticum%20aestivum;idx=;q=superoxide%20dismutase;site=ensembl ) . Are these 20 included in the 26 presented within the manuscript? etc. Please consider this topic, otherwise the manuscript appears ready for publication and should be consider for ‘minor revision’ before acceptance. This is almost ready. Congratulations on your contribution.

---

## Round 0.5 · accepted · Accept

· Academic Editor

Accept

Thank you addressing the comments related to the previous annotations of the IWGSC for the SOD genes. Your differences in approach did add some additional aspects to the description of the gene family and your justifications are acceptable. This probably reflects the automated processing that is often associated with a genome project versus the concentrated approach of a research group focusing on a key family of genes. I will grade this as accepted and pass the manuscript forward for final approval. Thank you for your contribution.